# Sitagliptin Is More Effective Than Gliclazide in Preventing  Pro-Fibrotic and Pro-Inflammatory Changes in a Rodent Model of Diet-Induced Non-Alcoholic Fatty Liver Disease

**DOI:** 10.3390/molecules27030727

**Published:** 2022-01-22

**Authors:** Jing Ren, Xiaoyu Wang, Christine Yee, Mark D. Gorrell, Susan V. McLennan, Stephen M. Twigg

**Affiliations:** 1Greg Brown Diabetes and Endocrinology Research Laboratories, Sydney Medical School (Central), Faculty of Medicine and Health, The University of Sydney, Camperdown, NSW 2006, Australia; renjing_alice@hotmail.com (J.R.); xiaoyu.wang@sydney.edu.au (X.W.); christine.yee@sydney.edu.au (C.Y.); susan.mclennan@health.nsw.gov.au (S.V.M.); 2Liver Enzymes in Metabolism and Inflammation Program, Centenary Institute, The University of Sydney, Newtown, NSW 2042, Australia; mark.gorrell@sydney.edu.au; 3A.W. Morrow Gastroenterology and Liver Centre, Royal Prince Alfred Hospital, Camperdown, NSW 2050, Australia; 4Department of Endocrinology, Royal Prince Alfred Hospital, Camperdown, NSW 2050, Australia; 5New South Wales Health Pathology (Eastern), Camperdown, NSW 2050, Australia

**Keywords:** NAFLD, sitagliptin, gliclazide, inflammation, fibrosis

## Abstract

A diet-induced non-alcoholic fatty liver disease (NAFLD) model causing obesity in rodents was used to examine whether sitagliptin and gliclazide therapies have similar protective effects on pathological liver change. Methods: Male mice were fed a high-fat diet (HFD) or standard chow (Chow) ad libitum for 25 weeks and randomly allocated to oral sitagliptin or gliclazide treatment for the final 10 weeks. Fasting blood glucose and circulating insulin were measured. Inflammatory and fibrotic liver markers were assessed by qPCR. The second messenger ERK and autophagy markers were examined by Western immunoblot. F4/80, collagens and CCN2 were assessed by immunohistochemistry (IHC). Results: At termination, HFD mice were obese, hyperinsulinemic and insulin-resistant but non-diabetic. The DPP4 inhibitor sitagliptin prevented intrahepatic induction of pro-fibrotic markers collagen-IV, collagen-VI, CCN2 and TGF-β1 and pro-inflammatory markers TNF-α and IL-1β more effectively than sulfonylurea gliclazide. By IHC, liver collagen-VI and CCN2 induction by HFD were inhibited only by sitagliptin. Sitagliptin had a greater ability than gliclazide to normalise ERK-protein liver dysregulation. Conclusion: These data indicate that sitagliptin, compared with gliclazide, exhibits greater inhibition of pro-fibrotic and pro-inflammatory changes in an HFD-induced NAFLD model. Sitagliptin therapy, even in the absence of diabetes, may have specific benefits in diet-induced NAFLD.

## 1. Introduction

Non-alcoholic fatty liver disease (NAFLD) is recognized as the most common chronic liver disease in developed countries [1]. The definition of NAFLD encompasses the entire spectrum of fatty liver disease, including simple steatosis and non-alcoholic steatohepatitis (NASH). At the NASH stage, liver steatosis, inflammation and hepatocyte injury (ballooning) with or without fibrosis are present, and NASH it at higher risk to progress to cirrhosis, liver failure and even hepatocellular carcinoma [2,3,4].

With the prevalence of obesity continuously increasing in western countries, the prevalence of NAFLD is showing similar trends [5]. NAFLD presence is estimated to be as high as 80% in obese patients, compared with 16% in individuals with normal BMI [6]. In obese people with NAFLD, excessive caloric intake is implicated in NAFLD development, so we adopted an established high-fat-fed mouse model to mimic these groups of patients. In a previous study of a high-fat-feeding mouse model, it was reported that after a prolonged dietary period of 20 weeks, the high-saturated-fat diet resulted in two of the three important features of NASH: lipid accumulation and leukocyte (macrophage) inflammatory infiltration, compared with chow-fed animals [7]

NASH is strongly linked to insulin resistance and type 2 diabetes [4,8]. However, data as to whether anti-diabetic agents or insulin sensitizers can help to attenuate NASH progression, especially in non-diabetic states, and related implicated mechanisms is limited. Sitagliptin is the dipeptidyl peptidase 4 (DPP-4) inhibitor that is most commonly prescribed to treat type 2 diabetes. Inhibiting the enzyme activity of DPP-4 lowers blood glucose because DPP-4 can rapidly proteolyse the incretin hormones glucagon-like peptide 1 (GLP-1) and glucose-dependent insulinotropic hormone (GIP) following their release in a meal-dependent manner. By increasing the levels of intact GLP-1, sitagliptin is able to increase insulin secretion and reduce glucagon release from the pancreas. Consequently, it improves hyperglycaemia [9,10]. In addition, DPP-4 is ubiquitous in both rodents and humans, as the liver one of the organs that most highly expresses DPP-4 [11]. Serum DPP-4 levels are increased significantly in NAFLD and NASH patients compared with controls [12,13]. The DPP-4 inhibitor, sitagliptin, has been demonstrated to attenuate steatosis in a diet-induced NAFLD model [14] and ameliorate NAFLD activity score (NAS) by improving steatosis and ballooning in NAFLD patients [15]. However, less evidence has been shown for the effect of sitagliptin on hepatic inflammation and fibrosis [16].

Another anti-diabetic agent used in this study is gliclazide, a second-generation sulphonylurea (SU) that is commonly used to treat type 2 diabetes. The drug acts by stimulating insulin release from pancreatic β-cells to lower blood glucose [17]. 

The aim of the current work was to examine whether DPP-4 inhibition in a diet-induced high-fat feeding NASH model shows efficacy in prevention of progressive NAFLD features at histological and gene-expression levels, compared with controls and sulphonylurea gliclazide treatment.

## 2. Results

### 2.1. Animal Profiles and Metabolic Measures

By week 25, as expected the HFD fed mice were on average 29% heavier than the standard Chow group (Figure 1A). The increasing body weight was not significantly affected by either sitagliptin or gliclazide (Figure 1A). The HFD fed mice, also as expected for this mouse strain, had normal blood glucose compared with the control group (Figure 1B). They had insulin resistance with a higher HOMA-IR value than Chow, and elevated circulating insulin (Figure 1C,D). Sitagliptin significantly improved the hyperinsulinemia and showed a trend to improve insulin resistance (Figure 1C,D). In contrast, also as expected in this high fat fed, non-diabetic model, gliclazide showed no effect on blood glucose, the circulating insulin level nor the HOMA-IR (Figure 1B–D). The insulin tolerance test (ITT) showed that the HFD group had a greater glucose area under the curve (AUC) value than Chow. Sitagliptin treatment showed a trend to normalize the AUC level, which was not significant (Appendix A). AST and ALT were measured and no obvious changes in HFD alone or HFD with sitagliptin treatment com-pared with Chow (Appendix A).

### 2.2. Sitagliptin Improves Steatosis and Inflammation 

Hepatic morphology assessed by H&E staining showed that most of the hepatocytes in the HFD-fed group contained numerous vacuoles, compared with control (Figure 2A,E vs. Figure 2B,F). Most of these vacuoles (Figure 2B,F) were of microsteatosis, while some of were of macrosteatosis in appearance. In addition, an inflammatory cell infiltrate was often observed around the central vein in the HFD group (Figure 2F, for example). Generally, across sections, as shown in the representative images, lipid accumulation induced by a high-fat diet was observed to be improved by sitagliptin, with a clear reduction in hepatocyte lipid droplet accumulation and lipid droplet size (Figure 2B vs. Figure 2C,F vs. Figure 2G in H&E; Figure 2J vs. Figure 2K,N vs. Figure 2O in Oil Red O stain). In contrast to effects of sitagliptin treatment in high-fat-fed mice, gliclazide treatment in high-fat-diet-fed mice did not show a clear reduction in steatosis compared with mice fed a high-fat diet alone (Figure 2B vs. Figure 2D,F vs. 2H in H&E; Figure 2J vs. Figure 2L,N vs. Figure 2P in Oil Red O stain).

Subsequently, gene expression of hepatic pro-inflammatory cytokines MCP-1, TNF-ɑ, IL-1β, and IL-6 was measured. A steady-state level of MCP-1 mRNA was significantly induced ~4.4 fold by HFD compared with control and was not normalized by either sitagliptin or gliclazide treatment (Figure 3A). TNF-α and IL-1β gene expressions were significantly increased to ~11 fold and ~4.8 fold, respectively, by HFD compared to chow and were significantly attenuated by sitagliptin treatment (Figure 3B,C). Liver IL-6 gene expression was markedly and significantly induced by HFD to ~6.9 fold, which appeared to show some inhibition by sitagliptin. However, this effect of the agent was not statistically significant (Figure 3D). In contrast to sitagliptin, gliclazide treatment did not significantly alter any of the liver mRNA levels of these pro-inflammatory cytokines when it was provided with the HFD (Figure 3A–D).

As a well-characterized mature macrophage marker [18], F4/80 was examined by immunostaining. F4/80 immuno-positive macrophages were shown to be statistically elevated in the HFD group compared with control mice (Figure 4A,B,E,F,I). This increase was partially inhibited by sitagliptin, but gliclazide treatment was without effect on macrophage infiltration (Figure 4C,D,G–I).

### 2.3. Sitagliptin Normalizes Fibrotic Markers

In examining the hepatic-fibrosis-associated collagens at the mRNA level, the HFD mice showed significantly enhanced gene expression of collagen-III and -IVα1, each compared with the control mice (Figure 5B,C). Collagen-VI a collagen involved in early liver remodelling, could not be reliably quantitated by PCR in the control group due to high cycle thresholds but could be assessed in HFD mice (Figure 5D). Type-I collagen was induced by HFD, but this was not statistically significant (Figure 5A). Sitagliptin treatment of HFD mice significantly prevented an increase in gene expression of collagen-IVα1 and -VI species compared with the HFD group (Figure 5C,D). However, gliclazide treatment did not prevent fibrotic gene induction of the liver (Figure 5A–D). In addition to collagen species, pro-fibrotic markers and potential mediators, cellular communication network factor 2 (CCN2) and transforming growth factor beta (TGF-β1) were found to be significantly induced by HFD (Figure 5E,F). These inductions were clearly prevented in each case by sitagliptin but not by gliclazide (Figure 5E,F).

Accumulation of crosslinked collagen in the liver was visualised by Sirius Red staining and immunohistochemistry. Lower and higher power images are shown in Figure 6A-L, and Figure 6a–l, respectively. HFD caused Sirius Red accumulation compared with control (Figure 6B,b,M). HFD also caused induction of type-I collagen protein (Figure 6F,f,N) and type-VI collagen protein (Figure 6J,j,O). In HFD mice, sitagliptin (Figure 6K,k,O), but not gliclazide treatment (Figure 6L,l,O), prevented the induction of collagen type-VI. Sitagliptin treatment showed a trend toward inhibition of Sirius Red induction (Figure 6C,c,M) and collagen type-I (Figure 6G,g,N), although in each case, this was not statistically significant.

CCN2 protein level was also examined by immunostaining. The intrahepatic CCN2 staining score was significantly elevated in the HFD mice compared with control (Figure 7A,B,E,F,I) and was significantly prevented by sitagliptin (Figure 7C,G,I). In contrast to sitagliptin, gliclazide had no effect on CCN2 liver protein scores (Figure 7D,H,I).

### 2.4. Exploration of Possible Underlying Mechanisms for Liver Protection by Sitagliptin 

Possible cellular pathways involved in the sitagliptin effect in HFD mice were examined through immunoblot analysis of whole-liver tissue lysate. Activation of the liver pro-fibrotic second messenger ERK pathway, detected as phospho-ERK, which was found to be potently increased in the HFD mice compared with controls, was markedly inhibited in sitagliptin-treated mice (Figure 8A,C). Notably, gliclazide-treated mice appeared to have some inhibition of ERK phosphorylation (Figure 8A), although this did not reach statistical significance (Figure 8C). In contrast to phospho-ERK, total ERK was not affected by HFD conditions or by the addition of sitagliptin or gliclazide to the diets (Figure 8A).

Dysregulation of autophagy is implicated in NAFLD progression [19]. Microtubule-associated protein 2 light chain 3 (LC3) and Beclin-1 are two markers used in this study to detect autophagy. LC3 is a marker of final autophagosome formation and Beclin-1 participates in the early stage of autophagy. HFD induced the liver-protein expression of LC3 II and Beclin-1 (Figure 8B,D,E). Both LC3 and Beclin-1 were downregulated by sitagliptin but not by gliclazide (Figure 8B,D,E).

Oxidative stress is another important pathogenic factor for NAFLD progression [20]. Malondialdehyde (MDA), an indicator of lipid peroxidation, and the activity of superoxide dismutase (SOD), an anti-oxidative enzyme, were measured in liver tissue to evaluate whether sitagliptin or gliclazide have any beneficial effect on oxidative stress. There were no significant changes in MDA content either in the HFD group or the HFD plus treatment groups compared with control (Figure 9A). Both total SOD activity (cytosolic + mitochondrial) and systolic SOD activity were reduced in the HFD group, but neither treatment normalized this impairment (Figure 9B,C).

## 3. Discussion

In this high-fat-diet model of obesity and insulin resistance without diabetes, we found that sitagliptin attenuates intrahepatic inflammation and fibrosis, possibly by reducing pro-fibrotic signalling via ERK and restoring the autophagic flux. The data show that sitagliptin has anti-inflammatory and anti-fibrotic effects for diet-induced NAFLD, compared with gliclazide, which is an agent mainly targeting blood-glucose control and with no such liver-protective effects.

In this study, the model reflected prediabetes conditions that are characterized by obesity, hyperinsulinemia, and a tendency toward hyperglycaemia. Long-term HFD consumption induces insulin resistance [21]. Sitagliptin-treated mice did not show weight loss in the obesity model, which is consistent with most clinical studies, where in the majority of people with diabetes, sitagliptin and gliclazide have a neutral effect on bodyweight regulation [22,23]. Thus, sitagliptin effects to target liver steatosis are not bodyweight-dependent in this model.

DPP-4 inhibitors are known to improve metabolic parameters and reduce plasma insulin levels in animal studies [24,25]. In the current study, the serum insulin level was decreased in HFD mice treated with sitagliptin, and HOMA-IR was partially normalized. In contrast, gliclazide did not influence the hyperinsulinemia in this HFD obese model.

Several recent studies have demonstrated that sitagliptin therapy decreases liver steatosis in diet-induced animal models with insulin resistance and metabolic syndrome [26] and also decreases steatosis and ballooning in NAFLD patients [15]. Selvihan et al. showed that sitagliptin may improve steatosis and hepatic insulin resistance through suppression of lipogenic and gluconeogenic pathways [14]. Concordantly, in our study, lesser lipid accumulation with macrovesicular steatosis was observed in the sitagliptin treatment group compared with HFD alone. In addition, in this current study, we report pro-inflammatory and pro-fibrotic data, with prevention of these pathological changes seen in NASH as a result of sitagliptin therapy in vivo.

NAFLD has been recognized as an inflammatory disorder [27], potentiated by chronic systemic inflammation, that is fundamental in the progression from simple fatty liver to the late stage of cirrhosis [28]. Excess hepatic lipid accumulation under inflammatory stress can trigger ER stress, oxidative stress and apoptosis, which may increase the risk of NASH-associated cirrhosis [29]. In our study, hepatic inflammation markers MCP-1 (CCL2), TNF-ɑ, IL-1β and IL-6 were induced in response to high-fat feeding. Sitagliptin inhibited increase in hepatic inflammatory markers in this high-fat-feeding-induced obesity model. Concordant with our results, Mona’s recent study demonstrated that sitagliptin suppresses inflammation by decreasing HMGB1-mediated TLR4/NF-κB signalling in a rat obesity model [30]. Similarly, Nagat reported anti-inflammatory action with significant reductions in inflammatory markers by DPP-4 inhibition in an NAFLD model of ovariectomized rats [31]. Moreover, DPP-4 deficient rats showed lower levels of hepatic pro-inflammatory and pro-fibrotic cytokines and reduced hepatic steatosis compared to wild-type rats [32]. 

The severity of liver fibrosis is regarded as the most important factor predicting NAFLD prognosis [33]. Patients determined by liver biopsy to have NASH and significant fibrosis have a higher risk for morbidity and mortality related to cirrhosis and liver complications compared to patients with simple steatosis [34]. In the current study, we first demonstrated that DPP-4 inhibitor prevents fibrosis development in the high-fat-diet-feeding-induced NAFLD mouse model. In several studies, DPP-4 is overexpressed or its activity is increased in fibrotic conditions. For example, it has been reported that serum DPP-4 activity is augmented in carbon tetrachloride (CCl4)-induced cirrhosis in rats [35]. Moreover, serum DPP-4 activity may be an indicator of the severity and progression of PBC [36]. Both DPP-4 knockout and DPP-4 inhibitor treatment in the CCl4-induced fibrosis model causes less intrahepatic crosslinked collagen accumulation and bridging fibrosis compared with control mice [37]. DPP-4 has the ability to bind to the extracellular matrix and is involved in hepatocyte-ECM interactions. High levels of DPP-4 activity may contribute to HSC-induced ECM accumulation in the liver [38,39]. In our in vivo study, ECM components, especially Col-IV and -VI, were downregulated by sitagliptin, possibly by inhibition of DPP-4 in the liver. Another explanation may be that sitagliptin protects hepatocytes by increasing the systemic GLP-1 level, which binds the GLP-1 receptor on hepatocytes, resulting in reduced hepatic insulin resistance and steatosis [32,40,41]. 

Several studies have reported that DPP-4 inhibition can suppress the activation of ERK1/2 [42,43]. Phosphorylation of ERK stimulated by TGF-β in skin fibroblasts can be attenuated by inhibitors of DPP-4 activity [42], and inhibition of ERK signaling pathways can decrease fibroblast proliferation [44]. In our study, the suppression of ERK phosphorylation by sitagliptin treatment provides a possible mechanism addressing how the anti-fibrotic effect in the NAFLD model is mediated. The inhibition effect on the ERK pathway by sitagliptin parallels effects seen on collagen and on the pro-fibrotic mediators CCN2 and TGF-β1.

The deregulation of autophagy is considered to play a crucial role in NAFLD progression [19]. Autophagy can regulate hepatocellular lipid accumulation by selective degradation. Thus, it has been implicated as playing a protective role in NAFLD. Blockade of autophagy flux (by blocking the fusion between autophagosomes and lysosomes) contributes to the development of hepatic steatosis [19]. Accumulation of LC3-II and p62 has been observed in NASH patients [45], and the accumulation of LC3 and p62 was positively correlated with disease severity [45,46]. However, autophagy may also be profibrogenic in liver fibrosis due to hepatic stellate cell activation [47]. The autophagy-associated protein LC3 was significantly elevated in the HFD group compared with the chow-fed group, which suggests that autophagic flux is impaired in the NAFLD model. Sitagliptin suppressed the expression of both autophagy markers, LC3 and Beclin-1, in the HFD group. This finding may provide a novel explanation for the protective effect of sitagliptin on NAFLD progression by restoring the autophagic flux.

Gliclazide is a second-generation sulfonylurea that, in diabetes, has shown greater safety, mild or neutral effects on body weight and protection against vascular complications, in addition to its effective blood glucose control [23]. However, in our obesity model with normal blood-glucose levels, gliclazide did not show any protective effects on NAFLD progression. These data suggest that this agent is less effective in NAFLD in the absence of diabetes.

In conclusion, sitagliptin protects against hepatic inflammation and fibrosis in diet-induced NAFLD with NASH, and this may occur by normalizing ERK-signaling and autophagy pathways, compared with gliclazide, which is mainly a hyperglycaemia control agent. This study broadens understanding of the effects of sitagliptin on NAFLD, in addition to the known protection against steatosis. Therefore, as sitagliptin does not generally cause hypoglycaemia in nondiabetic patients, sitagliptin may be a novel low-risk therapeutic option against hepatic inflammation and fibrosis for obese individuals with NAFLD. Whether these effects will occur in humans remains to be definitively reported in clinical trials. 

## 4. Materials and Methods

### 4.1. Animal and Experimental Design

Male C57BL/6 mice, aged 5 weeks, were purchased from Animal Resource Centre (Perth, Western Australia). They were maintained under controlled environmental conditions (12-h light, 12-h dark cycle and room temperature) and housed with access to food and water ad libitum. After 1 week of adaptation, they were randomly allocated into 2 groups and either fed standard chow (Chow; 12% kcal fat content, Specialty Feeds, Perth, WA, Australia) or a high-fat diet (HFD; containing 45% kcal fat content 20%, kcal protein and 35% kcal carbohydrate). The HFD diet was prepared in house, with a formula based on rodent research diet no. D12451 [48]. After 15 weeks of HFD, some mice from the HFD group were randomly selected and provided a diet containing a standard dose of oral sitagliptin (Januvia, Merck & Co, Kenilowrth, NJ, USA) (20 mg/day per mice) (3 g/kg of food) or gliclazide (DIAMICRON MR, Servier Laboratories, Sydney, NSW, Australia) (2 mg/day per mice) (0.3 g/kg of food) ad libitum, with doses following previous publications [49,50,51,52]. Animals were weighed, and blood glucose levels were measured weekly. Insulin tolerance tests were performed 1 week before termination [48]. HOMA index was calculated using the equation provided by Matthews et al. [53]. The Animal Ethics Committee of Sydney Local Health District, Sydney, Australia, approved all animal procedures. The number of animals in each group was (i) Chow, *n* = 5; (ii) HFD, *n* = 5; (iii) HFD + Sitagliptin, *n* = 8; (iv) HFD + Gliclazide, *n* = 7. Samples from all of the mice in each group were used to generate the quantitative data shown in each of the figures. 

### 4.2. Blood Sampling and Assays

At 25 weeks, blood was obtained by cardiac puncture under deep naesthesia using isoflurane. Serum was separated by centrifugation (3000 rpm, 10 min). Enzyme-linked immunosorbent assays for mouse insulin (EZRMI-13K; Millipore, Boston, MA, USA) were undertaken according to the manufacturer’s instructions. ALT activity was measured using an alanine aminotransferase activity assay kit (Sigma, St. Louis, MO, USA). Serum aspartate aminotransferase activity was measured using an aspartate aminotransferase (AST) activity assay kit (Sigma, St. Louis, MO, USA) according to the manufacturer’s instructions.

### 4.3. Quantitative Real-Time Polymerase Chain Reaction

Total RNA was extracted from liver tissue using the PureLink RNA Mini Kit (Life Technologies, New York, USA) according to the manufacturer’s instructions. Total RNA (2000 ng) from each sample was then reverse-transcribed to cDNA using SuperScript III reverse transcriptase, random hexamers and dNTP (Invitrogen, New York, USA). Expression of collagen-I, -III, -IV, -VI, CCN2, TGF-β, MCP-1, TNF-α, IL-1β, IL-6 and 18S as an internal control were determined by quantitative real-time PCR using SYBR green fluorophore (Invitrogen, New York, USA), as previously described [48]. The relative amount of each gene was determined by applying the threshold cycle to the standard curve. In each case, the expression level of the gene of interest was normalized to 18S ribosomal RNA and related to the relevant control, as previously described [54]. The primer sequences are shown in Table 1.

### 4.4. Tissue Preparation and Histological Studies, including Staining Scoring 

The left lobe of each liver was fixed in 10% buffered formalin, processed and embedded in paraffin for hematoxylin–eosin (H&E), Sirius red staining and immunohistochemistry staining for collagen-I, -VI and CCN2. The medial lobe of each liver was frozen in OCT. Frozen sections were stained with Oil-Red O to assess hepatic lipid content and F4/80 IHC. The primary antibodies were anti-collagen-I (1:400; AB765P; Millipore, Boston, MA, USA), anti-mouse collagen-VI (1:250; ab6588; Abcam, Cambridge, UK), anti-CCN2 (1:200; sc-14939; Santa Cruz Biotechnology, Santa Cruz, CA, USA) and anti-F4/80 (1:500; ab6640; Abcam, Cambridge, UK). Appropriate biotinylated secondary antibodies were used, and the signal was detected using Vectastain ABC kit (Vector Laboratories, Burlingame, CA, USA). Collagen by Sirius red stain, collagen-VI and CCN2 staining were scored for intensity by two independent scorers, each blinded to tissue and treatment group of origin, using a range between 0 and 4, as previously reported [48]. In brief, for each stain of interest, all sections were stained in the same batch. All stained sections were then photographed using an Olympus Provis AX70 (Olympus Optical Co. Japan) in 10 randomly chosen, standard-sized fields, which collectively covered most of each liver tissue section. Images were independently scored at 200x magnification. For each stain, all sections from all mouse groups were scored in the same session. The average of the scores was reported for each mouse studied. Then, group data were reported as mean ±SEM for the mice in each group. In reporting of results, scores were normalised to 1.0 for chow, and for fold-change reporting, were compared with chow in other groups.

### 4.5. Western Immunoblot 

Frozen liver tissue was homogenized in RIPA buffer containing protease (P8340; Sigma, St. Louis, MO, USA) and phosphatase inhibitors (201154; Sigma, St. Louis, MO, USA). Crude impurities were removed by brief centrifugation. Fat remaining on the tube wall was also carefully removed in order to prevent contamination of the homogenate. Total protein was separated in 10% SDS-PAGE gels and then transferred to PVDF membranes. The membranes were blocked with TBST containing 5% skim milk and then incubated with total-ERK antibody (1:500; 9102S; Cell Signaling, Danvers, MA, USA), phospho-ERK antibody (1:500; 9106S; Cell Signaling, Danvers, MA, USA), Beclin-1 antibody (Novus Biologicals, Centennial, CO, USA) and LC-3 antibody (Novus Biologicals, Centennial, CO, USA) in TBST containing 1% skim milk overnight at 4 ℃. After washing, the membranes were incubated with horseradish peroxidase-conjugated antibody (1:5000) for 2 h at room temperature. Bands were visualized using enhanced chemiluminescence (Amersham Biosciences, Piscataway, NJ). In each case, GAPDH (1:10,000; ab8245) or α-Tubulin (1:5000; ab7291; Abcam, Cambridge, UK) was detected and used as a loading control. Band intensity was quantitated using ImageLab.

### 4.6. Oxidative Stress Measurement

Malondialdehyde (MDA) content was measured using a thiobarbituric acid-reactive substances (TBARS) assay kit (Sigma-Aldrich, St. Louis, MO, USA) at 532 nm absorbance, and superoxide-dismutase (SOD) activity was measured using the xanthine oxidase method with an SOD kit (Sigma-Aldrich, St. Louis, MO, USA) at 550 nm absorbance, according to the manufacturer’s instructions.

### 4.7. Statistical Analysis

Results were analysed using Prism (GraphPad, San Diego, CA, USA) and shown as mean ± standard deviation (SD) or standard error of the mean (SEM). All data were compared using one-way ANOVA, followed by post comparisons using Bonferroni’s multiple comparisons test or unpaired *T*-test. Statistical significance was accepted at *p* < 0.05. 

## Figures and Tables

**Figure 1 molecules-27-00727-f001:**
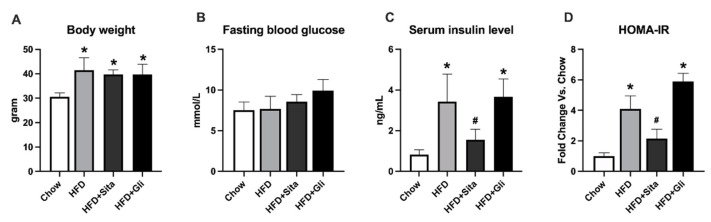
Animal characteristics and metabolic findings. Data are shown as mean ± SD at 25 weeks of standard chow or high-fat feeding, including the final 10 weeks with or without sitagliptin or gliclazide treatment. * *p* < 0.05 vs. chow alone; # *p* < 0.05 vs. HFD alone. (**A**) Body weight; (**B**) Fasting blood glucose; (**C**) Serum insulin level; and (**D**) Calculated HOM-IR, in respective groups as shown.

**Figure 2 molecules-27-00727-f002:**
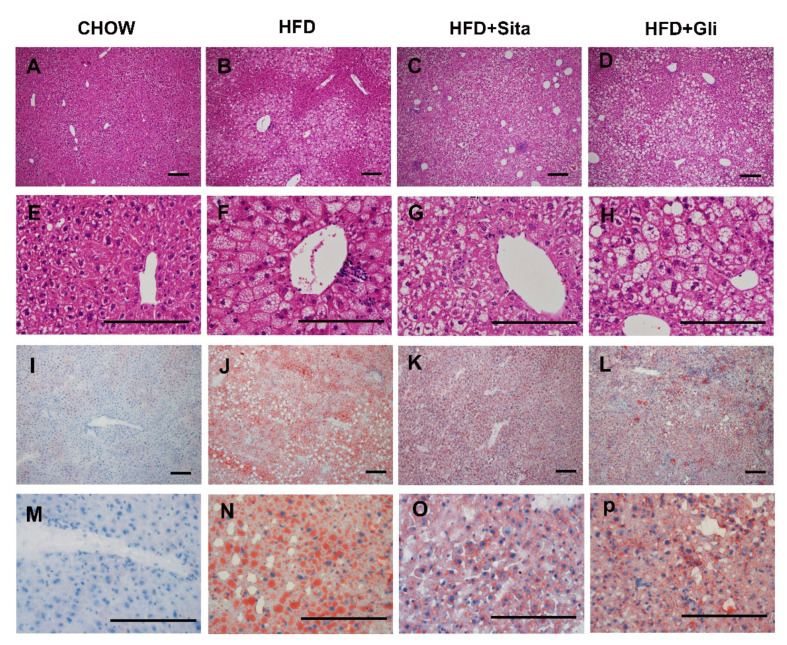
Histological change. Hematoxylin and eosin representative images at 100× (**A**–**D**) and 400× (**E**–**H**) magnification and representative images of Oil-Red-O-stained liver sections at 100× (**I**–**L**) and 400× (**M**–**P**) magnification. The scale bar is 200 microns.

**Figure 3 molecules-27-00727-f003:**
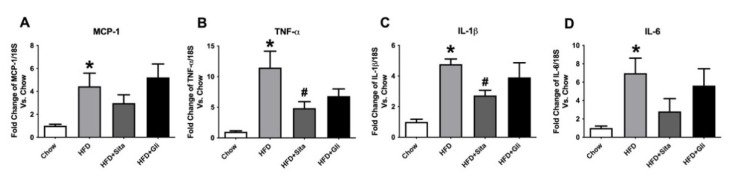
Hepatic gene expression of inflammation markers. MCP-1 (**A**), TNF-α (**B**), IL-1β (**C**) and IL-6 (**D**) mRNA quantitation by qPCR. Data are shown as Mean ± SEM of fold change compared with control mice. * *p* < 0.05, significantly different from chow alone; # *p* < 0.05, significantly different from HFD.

**Figure 4 molecules-27-00727-f004:**
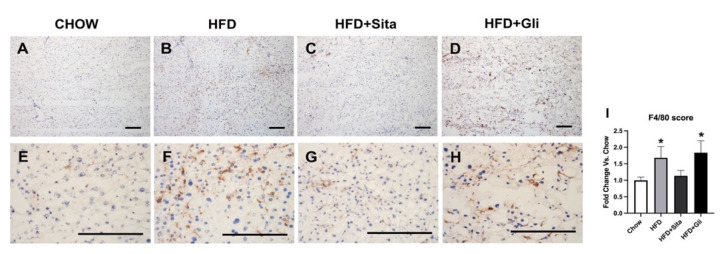
F4/80 immunohistochemistry. Representative images at 100× (**A**–**D**) and 400× (**E**–**H**) magnification. The scale bar is 200 microns. Results are shown as mean ± SD of fold change compared with staining intensity scores of control mice (**I**). * *p* < 0.05, significantly different from chow by unpaired *T*-test.

**Figure 5 molecules-27-00727-f005:**
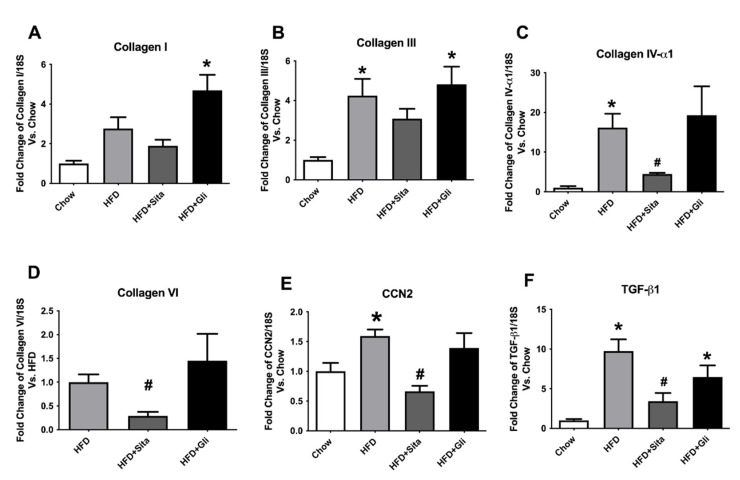
Hepatic gene expression of fibrosis markers: (**A**) Collagen-I; (**B**) collagen-III; (**C**) collagen IV–α1; (**D**) collagen VI; (**E**) CCN2 and (**F**) TGF-β1. mRNA quantitation by qPCR. Data are shown as mean ± SEM as fold change compared with control chow mice, (collagen-VI compared with HFD mice). * *p* < 0.05, significantly different from chow alone; # *p* < 0.05, significantly different from HFD.

**Figure 6 molecules-27-00727-f006:**
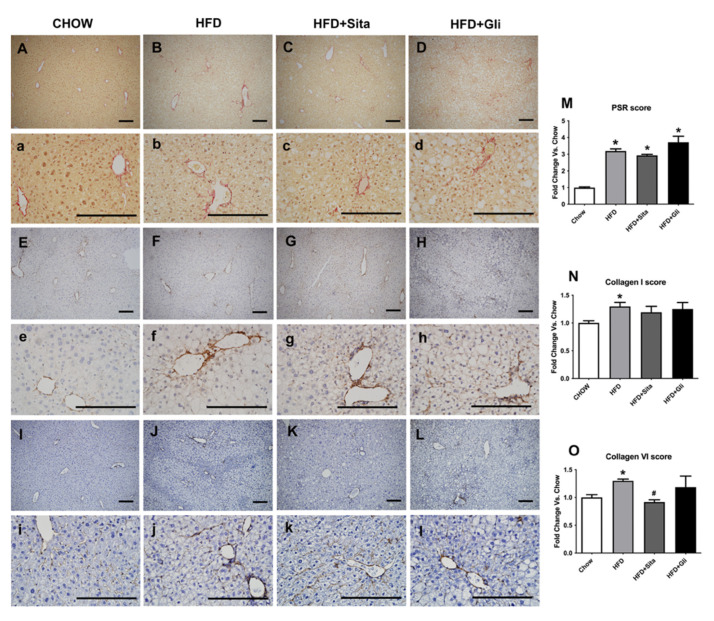
Liver Sirius red staining and type-I and Type-VI collagen immunohistochemistry. Representative images for Sirius red staining at 100× magnification (**A**–**D**) and at 400× magnification (**a**–**d**); for type-I collagen immunohistochemistry at 100× magnification (**E**–**H**) and at 400× magnification (**e**–**h**); and for type-VI collagen immunohistochemistry at 100× magnification (**I**–**L**) and at 400× magnification (**i**–**l**). The scale bar is 200 microns. Results are shown as mean ± SEM of fold change compared with staining intensity scores of controlled mice (**M**–**O**). * *p* < 0.05, significantly different from chow alone; # *p* < 0.05, significantly different from HFD, by unpaired *T*-test.

**Figure 7 molecules-27-00727-f007:**
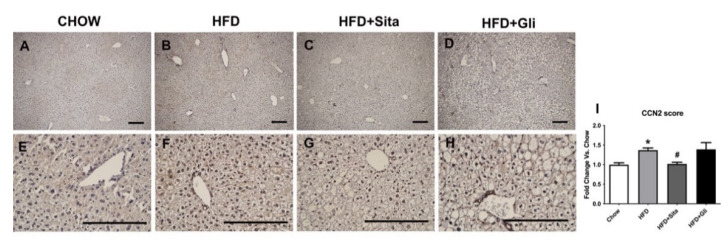
Liver CCN2 immunohistochemistry. Representative images at 100× (**A**–**D**) and 400× magnification (**E**–**H**). The scale bar is 200 microns. Results are shown as mean ± SEM of fold change compared with staining intensity scores of controlled mice (**I**). * *p* < 0.05, significantly different from chow alone; # *p* < 0.05, significantly different from HFD, by unpaired *T*-test.

**Figure 8 molecules-27-00727-f008:**
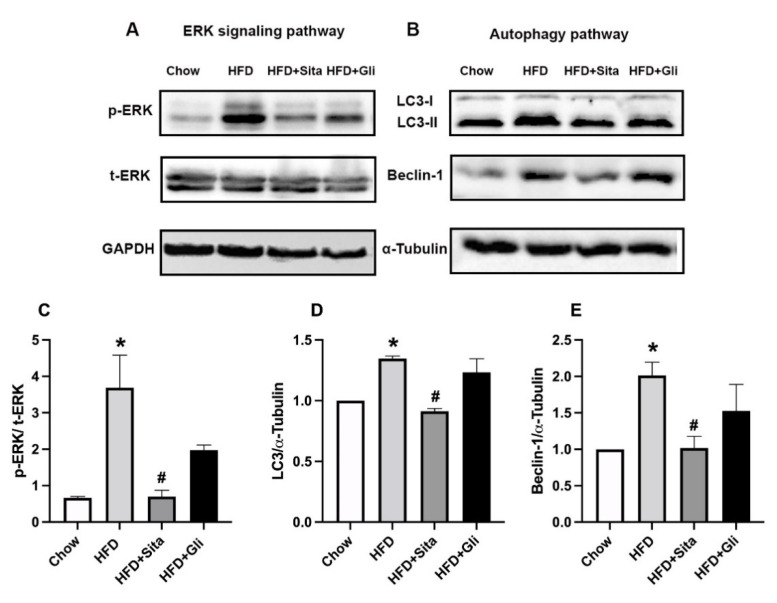
Effects of sitagliptin and gliclazide on ERK phosphorylation and autophagy activation. Phospho-ERK(p-ERK), total ERK(t-ERK), LC3 and Beclin-1 were detected by Western immunoblot with protein-loading control by GAPDH or ɑ-Tubulin. Representative images are shown in (**A**,**B**). Band intensity by Western immunoblot was quantitated by ImageLab. Data are shown as mean ± SEM of p-ERK, corrected by t-ERK (**C**), and as mean ± SEM of LC3 and Beclin-1 (**D**,**E**), corrected by corresponding loading control. * *p* < 0.05, significantly different from chow alone; # *p* < 0.05, significantly different from HFD.

**Figure 9 molecules-27-00727-f009:**
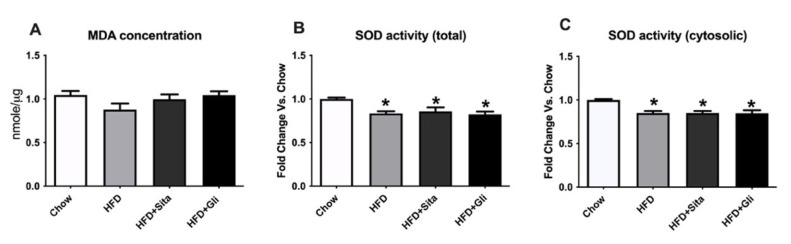
Effects of sitagliptin or gliclazide treatment on MDA content and SOD activity in high-fat-fed mice: (**A**) MDA content; (**B**) total SOD activity; (**C**) cytosolic SOD activity were measured. Data are shown as MDA concentration in Figure A and fold change of SOD activity as mean ± SEM in Figures B and C. * *p* < 0.05, significantly different from chow.

**Table 1 molecules-27-00727-t001:** Primer sequences used in the study.

	Forward	Reverse
**MCP-1**	**AGGTCCCTGTCATGCTTCTG**	**GCTGCTGGTGATCCTCTTGT**
**TNF-** **ɑ**	**CCCCAAAGGGATGAGAAGTT**	**CACTTGGTGGTTTGCTACGA**
**IL-1β**	**GACCTTCCAGGATGAGGACA**	**AGCTCATATGGGTCCGACAG**
**IL-6**	**TTCACAAGTCCGGAGAGGAG**	**TTCTGCAAGTGCATCATCGT**
**Collagen I**	**CCCCGGGACTCCTGGACTT**	**GCTCCGACACGCCCTCTCTC**
**Collagen III**	**CCTGGAGCCCCTGGACTAATAG**	**GCCCATTGCACCAGGTTCT**
**Collagen IV-** **ɑ** **1**	**ATCCGGCCCTTCATTAGC**	**ACTGCGGAATCTGAATGGTC**
**Collagen VI**	**GAACTTCCCTGCCAAACAGA**	**CACCTTGTGGAAGTTCTGCTC**
**CCN2**	**GAAGGGCAAAAAGTGCATCC**	**CAGTTGTAATGGCAGGCAC**
**TGF-β1**	**TGGAGCAACATGTGGAACTC**	**GTCAGCAGCCGGTTACCA**
**18S**	**CGGCTACCACATACCAAGGAA**	**GCTGGAATTAACCGCGGCT**

## Data Availability

Western immunoblot and SDS PAGE gel data were submitted with the manuscript. The total data can be made available upon reasonable request.

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
