# Peer review of "Sitagliptin Is More Effective Than Gliclazide in Preventing Pro-Fibrotic and Pro-Inflammatory Changes in a Rodent Model of Diet-Induced Non-Alcoholic Fatty Liver Disease"

_molecules, 2022, doi:10.3390/molecules27030727_

Round 1
Reviewer 1 Report
Sitagliptin is more effective than Gliclazide in Preventing Pro-Fibrotic 2 and Pro-Inflammatory Changes in a Rodent model of Diet-Induced 3 Non-Alcoholic Fatty Liver Disease
Comments:
Well designed and well-presented manuscript. I feel amendable deficiencies that are mandatory to be addressed before acceptance.
- Gliclazide is an antidiabetic drug but your findings are contraindicating. In figure 1 b, its noticeable that blood glucose is high in HFD+ Gliclazide group compared to HFD alone group please give reasoning for this contraindication, Similarly the circulating insulin levels and insulin resistance is significantly high in HFD+ Gliclazide group (figure 1 C and D) as appose to your description in the result section that Gliclazide has no effect circulating insulin and insulin resistance. So your figure is opposing your description.
- Histology images are of less resolution and clarity. We could not find much differences in histopathology of HFD and HFD+sitagliptin groups. Due to low resolution the hepatic structures are less demarcated. Please replace or improve the images so that the differences in morphology of different experimental groups should be clearly visible.
- Immunohistochemistry images are also of less resolution and cellular distribution of markers is not well characterized. Include images that will clearly show hepatocytes and visibly indicate the differences in experimental group. If the images will not show clear differences or clear ameliorating effect, then it goes opposite to the claims of the manuscript that Sitagliptin is more effected than HFD and Glicazide in improving fibrotic changes.
Regards
Reviewer 2 Report
This article is very interesting. Material and methods and results are correctly represented, discussion is adequate. Histological microphotographs are illustrative, but it is not clear how did you define staining intensity scores and levels 0-4 on histochemical and ICH staining. It should be more precise in Material and Methods.
I have to suggest that authors should put several recent references (within last 5 years).
